# Analysis of the Qualitative and Quantitative Content of the Phenolic Compounds of Selected Moss Species under NaCl Stress

**DOI:** 10.3390/molecules28041794

**Published:** 2023-02-14

**Authors:** Marija V. Ćosić, Danijela M. Mišić, Ksenija M. Jakovljević, Zlatko S. Giba, Aneta D. Sabovljević, Marko S. Sabovljević, Milorad M. Vujičić

**Affiliations:** 1Institute of Botany and Botanical Garden, Faculty of Biology, University of Belgrade, Takovska 43, 11000 Belgrade, Serbia; 2Institute for Biological Research “Siniša Stanković”, National Institute of the Republic of Serbia, University of Belgrade, Bulevar despota Stefana 142, 11060 Belgrade, Serbia; 3Department of Botany, Institute of Biology and Ecology, Faculty of Science, Pavol Jozef Šafárik University in Košice, Mánesova 23, 040 01 Košice, Slovakia

**Keywords:** bryophytes, in vitro culture, secondary metabolites, tolerance, HPLC, *Entosthodon hungaricus*, *Hennediella heimii*, *Physcomitrium patens*

## Abstract

The response to salt stress analysed by quantitative and qualitative analyses in three selected moss species was studied. Non-halophytic funaroid *Physcomitrium patens* and two halophytic mosses, funaroid *Entosthodon hungaricus* and pottioid *Hennediella heimii* were exposed to salt stress under controlled in vitro conditions. The results clearly showed various phenolics to be present and included to some extent as a non-enzymatic component of oxidative, i.e., salt stress. The common pattern of responses characteristic of phenolic compounds was not present in these moss species, but in all three species the role of phenolics to stress tolerance was documented. The phenolic p-coumaric acid detected in all three species is assumed to be a common phenolic included in the antioxidative response and salt-stress tolerance. Although the stress response in each species also included other phenolics, the mechanisms were different, and also dependent on the stress intensity and duration.

## 1. Introduction

Bryophytes (liverworts, hornworts, and mosses) are the second largest group of terrestrial plants [1] inhabiting numerous habitat types (absent only from seas), due to their great adaptability to various environmental conditions. However, although they are not present in sea waters, there are salt-tolerant bryophyte representatives which inhabit salty environments (e.g., moss *Entosthodon hungaricus* (Boros) Loeske from salty grassland, liverwort *Riella echinata* (Müll. Frib.) Segarra, Puche and Sabovlj. from brackish inland waters stands and many others) [2].

At present, soil salinisation (mainly by sodium chloride) is one of the major abiotic stress factors which harm and/or restrict plant growth and reduce plant productivity [3]. Despite being poorly competitive with vascular plants, bryophytes have a plethora of survival strategies enabling them to cope with harsh environments including salt stress. In general, salt-tolerant bryophytes, i.e., bryo-halophytes [2,4,5], often tolerate high salt concentrations by engaging numerous molecular, physiological, ecological, and biochemical mechanisms [2,6,7,8]. Unfortunately, many of these mechanisms remain obscure and differ among species. In addition to exposure to salt stress, and experiencing suboptimal functional stages, bryophytes also have to cope with the oxidative stress which arises as the consequence of the high production of reactive oxygen species (ROS) and the decreased ability of antioxidative mechanisms to keep the ROS level low [9]. Nevertheless, it is assumed that bryophytes are well adapted to oxidative stress [10] due to the presence of many biologically active compounds in their bodies such as mono-, di-, tri- and sesquiterpenoids, bibenzyls, phenolics/flavonoids, dihydrostelbenes, alkaloids, glycosides, saponins, anthraquinones, sterols and other aromatic compounds [11,12,13,14]. Many of the aforementioned metabolites are mostly detected and described in the liverwort division because they possess specific oil bodies in fresh plant material in which chemically interesting lipid and terpenoid compounds are accumulated [15,16]. On the other hand, in the moss division, many secondary metabolites are present in the specific cell compartments noticed in vacuoles or as independent lipid droplets [17,18], many of which may serve as potent antioxidants and contribute to oxidative stress resistance. Such chemical diversity may be one of the key factors promoting strong protective mechanisms for the survival of extreme environmental conditions.

Among all the mentioned metabolites, phenolics are the dominant compounds of the studied moss species. Although phenols are numerous and diverse in bryophytes and other plants (with over 8000 compounds known so far; [19]), their specific roles and functions have not been completely elucidated [20,21]. Nevertheless, oxidative stress protection is one of the most studied functions of phenols and according to Sgherri et al. [22], phenolics are electron donors for vacuolar peroxidases or activators of other antioxidative enzymes and thus contribute to the considerable antioxidative capacity of bryophytes [23]. 

In general, four main classes of phenolics can be distinguished, i.e., phenolic acids, flavonoids, stilbenes, and lignans [24,25]. Additionally, coumarins and tannins can also be distinguished, according to Shahidi and Yeo [19]. Their content and concentrations are often *species*-specific and also depend on stress intensity, duration, and environmental conditions. Low or moderate salinity can elevate the phenolic content in mosses [26], but a decrease in the total phenolics can also occur [27]. In addition to phenolics, other protective metabolites such as ascorbate, glutathione, tocopherols, polyols, and carotenoids also play an important role in antioxidative protection alongside antioxidative enzymes (peroxidases, catalases, and superoxid dismutases). Those two components, i.e., enzymatic and non-enzymatic, both participate in salt and oxidative stress tolerance [28].

Despite the large amount of new information pertaining to bryophyte resistance and tolerance mechanisms in the last decade, many problems linked to this phenomenon remain to be resolved since some of the mechanisms of these peculiar plants can be used in crop improvements. Therefore, the aims of this research were as follows: (1) to investigate how NaCl affects the total phenolic content in selected mosses (non-halophyte *Physcomitrium patens* (Hedw.)) Mitt. and two bryo-halophytes *Entosthodon hungaricus* (Boros) Loeske and *Hennediella heimii* (Hedw.) R. H. Zander); (2) to explore how concentrations of individual phenolic compounds change during short-term and long-term NaCl stress; (3) to discover the differences in the survival strategies of the selected moss species, if any; and (4) to point out the common and specific phenolic compounds which play a key role in the response to NaCl in any of the species studied. 

## 2. Results

All the tested moss species had normal gametophore development and the effects on the development and morpho-anatomical features were elaborated elsewhere [29,30].

Among the tested moss species, non-halophyte *P. patens* had the greatest amount of total phenolic content measured by the Folin-Ciocalteu method in the control plants (Figure 1A). The control plants, as well as the plants grown on medium with low concentrations of NaCl under long-term stress (0–10 mM), produced higher amounts of total phenolics than salt-stressed plants for three days only. However, when the concentrations of NaCl were increased (50 and 100 mM), the number of total phenolics also increased in those plants exposed to salt for three days, but decreased in the plants grown in medium containing NaCl for three weeks, indicating different responses to salt stress related to exposure. Interestingly, when the plants were exposed to higher NaCl concentrations (200–500 mM), the response was similar for both short-term and long-term stress, i.e., the total phenolic content rapidly decreased compared to the control samples. The plants grown on medium supplemented with a rather high concentration of 500 mM of NaCl also produced phenolics, albeit in lower amounts, indicating that phenolics jointly with some other compounds/ROS regulation mechanisms play a role in surviving such harsh conditions in this non-halophytic moss.

Compared to *P. patens*, facultative halophyte *E. hungaricus* [31] displays a different response to NaCl (Figure 2A). The greatest amount of total phenolics was observed in the control plants of both short- and long-term stress. When NaCl was applied, the total phenolic content decreased linearly with the increase in the concentrations of NaCl. Although *E. hungaricus* is a halophytic moss, high concentrations of NaCl (500 mM) greatly affected the content of total phenolics, suggesting their significant role in surviving high salt concentrations, but also inferring that some other mechanisms support its salt-stress survival.

A similar phenolic content pattern to that shown in *E. hungaricus* was observed for bryo-halophyte *H. heimii* (Figure 3A). Nevertheless, the control plants had somewhat lower phenolic content than the other two studied mosses. The total phenolics decreased linearly with the increment of NaCl concentrations both in short- and long-term stress. However, significant amounts of total phenolics were detected in the plants grown on a medium supplemented with high concentrations of NaCl (200–500 mM), suggesting the presence of different salt tolerance mechanisms and survival strategies in *E. hungaricus*.

In general, the results obtained with HPLC (high-performance liquid chromatography) analysis showed a greater diversity of phenols in the tested mosses exposed to short-term NaCl stress than those exposed to long-term stress. In total, nine different phenols were detected in all three tested moss species, although their concentrations greatly varied depending on the NaCl concentration and duration of the salt stress. Thus, two phenolic acids (p-coumaric acid and caffeic acid) and seven flavonoids from different classes, i.e., flavons (apigetrin and apigenin), flavonols (rutin, isoquercitrin, astragalin and kaempferol) and flavanone (naringenin) were detected.

Non-halophytic moss *P. patens* had the greatest concentration of total phenolics among all the tested species, with the dominant presence of p-coumaric acid, especially in the plants exposed to long-term stress (Figure 1B). The concentrations of the other detected phenols were rather low in the plants subjected to prolonged exposure to NaCl. In contrast, when NaCl was applied for three days, the presence of kaempferol, astragalin and isoquercetin was evidently higher (Figure 1B,D). Additionally, in the plants exposed to extremely high NaCl concentrations (500 mM) there was a significant increase in the concentrations of caffeic acid which was not detected in the other experimental groups or was weakly present. On the other hand, caffeic acid was permanently present in the plants during long-term stress in relatively small amounts (Figure 1B). These results suggest a rather high diversity of phenols in *P. patens*, some of which decrease salinity stress. It can thus be inferred by the quantitative analyses that p-coumaric acid may play a key role among phenol components in the response to increased NaCl concentrations.

As was the case in *P. patens*, p-coumaric acid was the dominant phenolic acid during salt stress in bryo-halophyte *E. hungaricus* (Figure 2B). These two species are phylogenetically closer, funaroid mosses, and belong to the same Funariaceae family. Nevertheless, the concentrations, as well as the diversity of the identified phenolics, were lower than those in *P. patens*, which also resulted in lower total phenolic contents (Figure 2A). In the plants exposed to low NaCl under short-term stress, rutin, astragalin and apigenin were present, whereas the concentration of kaempferol increased in those plants exposed to 300 and 500 mM of NaCl. Interestingly, kaempferol was not detected at all in long-term NaCl stress (Figure 2B,D). The relative presence of specific phenols indicates that p-coumaric acid is the dominant phenolic acid playing the main role in the response of *E*. *hungaricus*, especially in plants exposed to long-term stress.

In comparison to the funaroid mosses, bryo-halophyte *H. heimii* (Pottiaceae) exhibited the lowest concentration of both phenolic acids and flavonoids. However, huge diversity in the phenolic compounds present was evident (Figure 3B). Interestingly, p-coumaric acid was not present in large amounts in the *H. heimii* control plants, but with the application of increasing NaCl concentrations, its contribution to the total phenolic content also rose (Figure 3B). These results undoubtedly indicate that p-coumaric acid has the dominant role among phenolics in the response to NaCl stress in *H. heimii*. The concentrations of the detected phenolic compounds varied in a specific manner, i.e., rutin and astragalin decreased in short-term stress, while the concentrations of isoquercitrin, p-coumaric acid, apigenin, naringenin and kaempferol significantly increased (Figure 3C). The concentrations varied greatly among the plants exposed to different NaCl concentrations for three weeks with the exception of caffeic acid, which was only present in the plants exposed to low concentrations of NaCl in long-term stress (Figure 3B,D).

## 3. Discussion

The synthesis and accumulation of phenolic compounds can be induced endogenously during plant development, but also due to environmental effects, i.e., salt and/or oxidative stress [32]. In optimal conditions, the antioxidative protective system (composed of enzymes and non-enzymatic compounds) provides adequate protection from ROS present in the cells, maintaining their concentration at a low level [33]. However, when plants are exposed to increased salinity, and thus increased oxidative stress, phenolics can donate protons or electrons to antioxidative enzymes and neutralise free radical species [24]. In this study, the total phenolic content varied among the three investigated species and depended on the duration of stress (short-term or long-term stress). Despite being evolutionarily close, the funaroid mosses, *P. patens* and *E. hungaricus,* did not share a similar trend in total phenolic changes in short-term NaCl stress (Figure 1A and Figure 2A). However, a corresponding linear decrease in the total phenolic compounds was evident for phylogenetically unrelated bryo-halophytes, *E. hungaricus* and *H. heimii* (Figure 2A and Figure 3A), indicating a similar mechanism used by halophytic bryophytes in short-term exposure to NaCl. Conversely, when the plants were grown on media supplemented with NaCl for three weeks (long-term stress), a different pattern was observed, although the total phenolic content decreased in all three studied species. In bryo-halophyte *E. hungaricus* the total phenolics significantly reduced when the plants were exposed to extremely high concentrations of NaCl, suggesting that phenolics are among the main components which contribute to salt tolerance due to their consumption and low concentration. In the other two tested species, the total phenolics decreased moderately, which indicates the involvement of some other protective components besides phenolics in salt stress. These suggest bryo-halophytes, namely *E. hungaricus* and *H. heimii,* have different strategies for salt-stress response and tolerance. This has already been shown in some other responses in these two species [29,30]. 

Salt stress can lead to either increasing or decreasing total phenolic content in plants. The synthesis of phenols is often noticeable in plants exposed to low or moderate salinity, while in extreme conditions the concentration of total phenols often decreases [26,34]. In bryophytes, the effects of NaCl on total phenolics are diverse, but high NaCl concentrations often lead to the decreasing of phenol concentrations, as also shown for *Bryum argenteum* Hedw. and *Atrichum undulatum* (Hedw.) P. Beauv. [27]. The tested mosses in this study showed a similar pattern in total phenolic content when high NaCl concentrations were applied, suggesting that phenolics are included in antioxidative protection. In general, the concentration of phenolics accumulated in plants depends on plant species, duration of stress, the concentration of NaCl, and the developmental phase [35,36,37]. We assume that accession, i.e., geotype, genotype or even ecotype, can also trigger a different response. It should be taken into account that in real contexts, i.e., the varying habitat types, even the microhabitats of each of the tested species, and other ecological conditions can also affect salt tolerance, synergistically or antagonistically. Thus, tolerance can vary depending on the situation, i.e., a combination of environmental parameters, as previously stated [2]. 

Bryophytes are very rich in secondary metabolites, especially phenolics. Although their chemical diversity has been studied to a certain extent, significantly less is known about their specific function and role in salt stress (e.g., [20,21]). In this study, nine different phenolic compounds were detected. Interestingly, the dominant phenol in all three investigated species was p-coumaric acid, mostly present in all the experimental groups. Bearing in mind that p-coumaric acid is a precursor for phenolic acid and flavonoid biosynthesis [19,38], those results were not completely unexpected. The production of phenolics is initiated via the phenylpropanoid pathway in all plants where all compounds are derived from p-coumaric acid through the action of different enzymes [39]. An increase in the concentration of p-coumaric acid was present in all the tested species growing in moderate salinity both in long-term and short-term stress, which suggests the importance of this specific phenolic acid in the response to NaCl. According to Erxleben et al. [40], caffeic acid and chlorogenic acid were the dominant phenolic acids in non-stressed gametophores of *P. patens*. Conversely, in long-term stress when extreme NaCl concentrations were applied, the concentration of p-coumaric acid decreased in all the tested species, but in a *species*-specific manner. 

Some other interesting phenolic acids have been previously detected in different bryophyte clades, such as methyl p-coumaric acid, caffeic acid methyl ester, and rosmarinic acid in hornworts [41]. In mosses, caffeic acid has been reported in various representatives (also in pleurocarp *Brachytheciastrum velutinum* (Hedw.) Ignatov and Huttunen, and *Kindbergia praelonga* (Hedw.) Ochyra [42].

On the other hand, the flavonoid content mostly decreased in the plants exposed to NaCl treatments. However, in bryo-halophyte *H. heimii*, the presence of flavonoids was evident both in short-term and long-term stress (Figure 3B) compared to the funaroid mosses (*P. patens* and *E. hungaricus*) where phenolic acids were the dominant phenolic components (Figure 1B and Figure 2B). A greater presence of flavonoids than of phenolic acids was found in the bryo-halophytes exposed to short-term stress suggesting their role in plant protection from short-term exposure to NaCl which is not usually very harmful to halophytes. Nevertheless, non-halophytic moss *P. patens* also accumulated flavonoids apigenin and kaempferol under short-term stress, but in very low concentrations (Figure 1B). Interestingly, bryo-halophyte *H. heimii* had the lowest concentration of total phenolic content, but more flavonoids than the other two investigated species. In moderate and extreme long-term stress, besides the dominant p-coumaric acid, *H. heimii* also contained high concentrations of isoquercitrin, naringenin, apigenin, and kaempferol (Figure 3B). The flavonoid naringenin is a precursor for flavanone and flavonol synthesis [19,38], and therefore any decrease or increase in concentration is closely linked to the concentration of subsequent flavonoids such as apigetrin, apigenin, rutin, isoquercitrin, and kaempferol. Increased naringenin concentrations under moderate stress can be explained as the accumulation of naringenin due to reduced concentrations in the synthesis of other flavanones and flavanols which was detected in *H. heimii*. In general, *H. heimii* showed a different metabolic pattern compared to the funaroid mosses, where flavonoids accumulated more often than phenolic acids. Interestingly, moss *P. patens* is more plentiful in phenolic acids than in flavonoids when compared to bryo-halophytes, which may suggest the ability of p-coumaric acid to contribute to the greater tolerance to NaCl stress of such a non-halophyte, and that although this species does not inhabit salt stands, it has some evolutionarily memorised pathway to cope with certain salt stress as previously suggested by Frank et al. [43]. Flavonoids such as flavones, flavonols, isoflavones, dihydroflavonols, and aurones were detected in bryophytes with the dominant record of luteolin and quercetin [44]. Compared to liverworts, mosses are less studied. However, in *Brachytheciastrum velutinum* and *Kindbergia praelonga* flavonoids apigenin-7-O-glucoside, luteolin and apigenin were detected [42]. Moreover, biflavonoids and triflavonoids are typically present in mosses compared to liverworts where monoflavonoids are more likely to be found [44]. Interestingly, hornworts seem not to synthesise flavonoids [45,46]. In addition to this great variety of flavonoids, bryophytes, mainly liverworts, also accumulate phenolics in multimeric forms such as bibenzyls, bibenzyls and bisbibenzyls [44,47].

This huge chemical diversity has often been studied as the potential for antimicrobial and antibacterial protection, not as abiotic stress-protective compounds. However, such information is of great importance since plants are not known to possess enzymes which neutralise the hydroxyl radical (·OH), the most reactive form of ROS. Therefore, non-enzymatic components, such as phenolics, can effectively react with ·OH and protect cells from oxidative damage [48]. Moreover, flavonoids can react with ^1^O_2_ (singlet oxygen) and thus mitigate damage in the outer chloroplast membrane [49].

Based on the novel results presented in this study and the previous knowledge on salt stress in bryophytes, it can be inferred that although the three tested moss species differ in their reaction to salt stress, phenolic compounds are included in oxidative stress responses. This was demonstrated in both halophytic and non-halophytic species. It can be assumed that the mechanisms of stress tolerance are dissimilar among the tested species.

## 4. Materials and Methods

Bryophytes, including mosses, are rarely tested for salt stress since they do not live in salt water. However, there are numerous species which have to cope with salt stress or even live in such environments as brackish water. With the aim of studying the response of mosses to salt and oxidative stress, we chose two phylogenetically unrelated species of bryo-halophytes, namely pottioid *Hennediella heimii* and funaroid *Entosthodon hungaricus* and a model moss *Physcomitrium patens*. For details on the already known biology of the species, please refer to Sabovljević et al. [5,31] and Ćosić et al. [50], while information on the axenic cultivation of bryophytes can be found in Sabovljević et al. [51].

### 4.1. Plant Material and Experimental Design

Axenic in vitro cultures of the experimental mosses, *P. patens*, *E. hungaricus* and *H. heimii,* were established and the full development of gametophores was achieved. Details of the culture procedure can be found in Sabovljević et al. [5,31] and Ćosić et al. [50]. Having achieved a sufficient number of developed gametophores to start the experiments, cultures of the selected mosses were grown on solid BCD medium (containing 0.2 M MgSO_4_ × 7H_2_O, 0.18 M KH_2_PO_4_, 1 M KNO_3_, and 0.9 mM FeSO_4_ × 7H_2_O) with sucrose added (0.05 M) to achieve quick growth and morphologically well-developed gametophytes [5,52] prior to the experimental tests. The pH of the media was adjusted to 5.8 before autoclaving at 121 °C for 30 min. Further tests were conducted on minimal BCD medium supplemented with different NaCl concentrations before autoclaving (5, 10, 50, 100, 200, 300, and 500 mM NaCl). In experiment type I, the mosses were grown on BCD medium containing NaCl for 3 days, which simulated short-term stress. Experiment type II represented long-term stress, where the mosses were grown on BCD medium containing NaCl for 21 days. The control plants were grown on BCD salt-free medium. The culture conditions were set at 18 ± 2 °C, under a long-day photoperiod (16 h:8 h light/dark). All the plants tested here were grown under the same controlled laboratory conditions including the light quality and intensity (47 μmol m^−2^s^−1^ irradiance). After 3 and 21 days, respectively, the plant material was collected and frozen at −70 °C until further analysis. All the chemicals were supplied by Sigma Aldrich, Germany.

### 4.2. Quantification of the Total Phenolic Content

The total phenols were extracted in 96% methanol (*v:v*) from 200 mg of frozen plant material and incubated for 60 min at room temperature. After incubation and centrifugation at 12,000× *g* for 15 min, supernatant was used for the analysis of the total phenols according to the Singleton and Rossi method [53]. The Folin-Ciocalteu reagent was added to the supernatant of each sample and incubated at room temperature for 6 min after which Na_2_CO_3_ was added to the mixture (7.5% solution). After 2-h incubation in the dark, absorbance was measured at 740 nm using a Multiskan Sky Thermo Scientific microtiter plate reader (Finland). The total phenols were calculated from gallic acid (GAE) calibration curve (0–1000 µg mL^−1^) and expressed as mmol GAE per g dry weight.

### 4.3. UHPLC/-HESI-MS/MS Analysis of the Phenolic Compounds

The separation, identification, and quantification of the components in the methanol extracts of gametophytes and sporophytes was performed using the Dionex Ultimate 3000 UHPLC system (Thermo Fisher Scientific, Bremen, Germany) connected to a TSQ Quantum Access Maxtriple-quadrupole mass spectrometer (Thermo Fisher Scientific, Basel, Switzerland). Elution was performed at 30 °C on a Hypersil gold C18 column (50 × 2.1 mm), with 1.9 µm particle size (Thermo Fisher Scientific, Waltham, MA, USA). The mobile phase consisted of (A) 0.01% acetic acid and (B) acetonitrile (MS grade, Fisher Scientific, Loughborough, UK), which were applied in the following gradient elution: 5–20% B in the first 3.0 min, 20–40% B 3.0–5.0 min, 40–50% B 5.0–7.5 min, 50–60% B 7.5–8.5 min, 60–95% 8.5–10.5 min, from 95% to 5% B until 12th min and 5% B until 15th min. The flow rate was set to 0.4 mL min^−1^ and the detection wavelengths to 260 and 320 nm. The injection volume was 15 μL. All the analyses were performed in triplicate.

A TSQ Quantum Access Max triple-quadrupole mass spectrometer equipped with heated electrospray ionization (HESI) source was used with the vaporising temperature maintained at 450 °C, and ion source settings described by Mišić et al. [54]. The mass spectrometry data were acquired in negative mode, in m/z range from 100 to 1000. Collision-induced fragmentation experiments were performed using argon, with collision energy set to 30 eV. A time-selected reaction monitoring (tSRM) experiment for the quantitative analysis of nine phenolic compounds was performed by using two MS^2^ fragments for each compound. Xcalibur software (version 2.2) was used for instrument control, data acquisition, and analysis.

The phenolic acids and flavonoids were identified by direct comparison with commercial standards: caffeic acid (CA), p-coumaric acid (p-CA), rutin (Ru), isoquercitrin (Iq), astragalin (As), apigetrin (Ap), naringenin (Na), apigenin (Ag), and kaempferol (Ka) (Sigma Aldrich, Germany). The total amount of each targeted compound in the gametophyte material was calculated based on a calibration curve of pure compound (1 ng mL^−1^ to 50 µg mL^−1^) and expressed as ng per 100 mg fresh weight. 

### 4.4. Statistical Analysis

For each NaCl treatment and each tested species, there were 4 replications (Petri dishes) containing 10 gametophore patches of the same size (5 mm in diameter). The collected data were analysed using Statistica software version 8 (STSC Inc., Rockville, MD, USA), followed by ANOVA analysis of variance. Differences were considered significant for values of *p* < 0.05.

## 5. Conclusions

The qualitative and quantitative analyses of the phenolic content in the three tested mosses show differences depending on species, but also on exogenous factors and the intensity of stress, as well as the exposure time to stressful conditions. The obtained results clearly demonstrate that phenolics are included in salt and oxidative stress responses both in halophytic and non-halophytic mosses and also suggest that the mechanisms differ among the tested mosses.

## Figures and Tables

**Figure 1 molecules-28-01794-f001:**
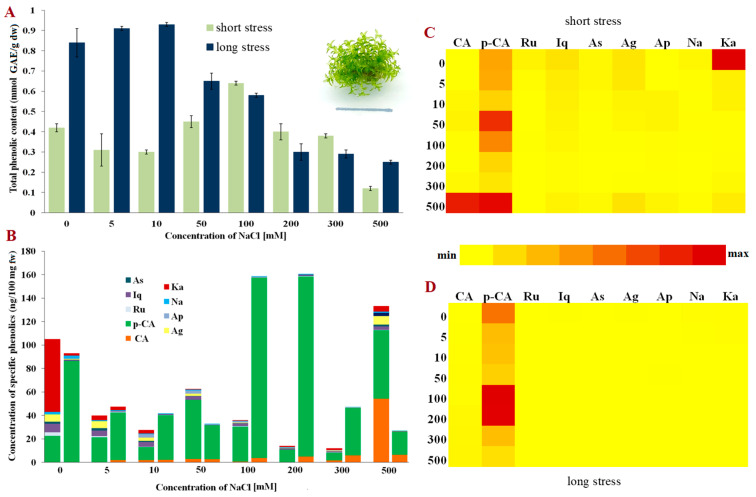
Phenolic content in *Physcomitrium patens*. (**A**) Total phenolic content in short- and long-term stress; (**B**) Content of phenolic compounds; (**C**) Relative representation of phenolic compounds in short-term stress; (**D**) Relative representation of phenolic compounds in long-term stress. Abbreviations: astragalin (As), isoquercitrin (Iq), rutin (Ru), p-coumaric acid (p-CA), caffeic acid (CA), kaempferol (Ka), naringenin (Na), apigetrin (Ap), apigenin (Ag), gallic acid (GAE), fresh weight (fw) and dry weight (dw).

**Figure 2 molecules-28-01794-f002:**
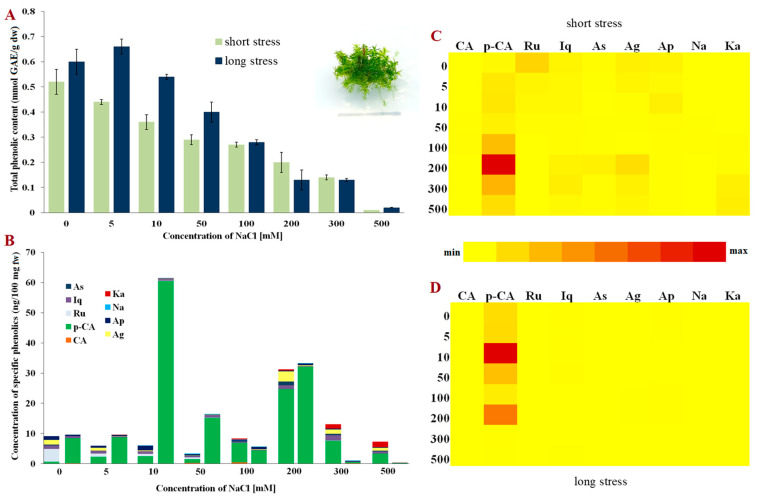
Phenolic content in *Entosthodon hungaricus*. (**A**) Total phenolic content in short- and long-term stress; (**B**) Content of phenolic compounds; (**C**) Relative representation of phenolic compounds in short-term stress; (**D**) Relative representation of phenolic compounds in long-term stress. Abbreviations: astragalin (As), isoquercitrin (Iq), rutin (Ru), p-coumaric acid (p-CA), caffeic acid (CA), kaempferol (Ka), naringenin (Na), apigetrin (Ap), apigenin (Ag), gallic acid (GAE), fresh weight (fw) and dry weight (dw).

**Figure 3 molecules-28-01794-f003:**
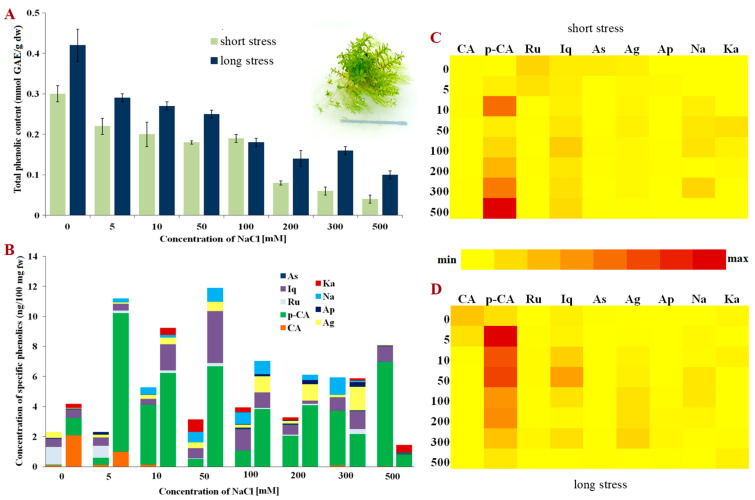
Phenolic content in *Hennediella heimii*. (**A**) Total phenolic content in short- and long-term stress; (**B**) Content of phenolic compounds; (**C**) Relative representation of phenolic compounds in short-term stress; (**D**) Relative representation of phenolic compounds in long-term stress. Abbreviations: astragalin (As), isoquercitrin (Iq), rutin (Ru), p-coumaric acid (p-CA), caffeic acid (CA), kaempferol (Ka), naringenin (Na), apigetrin (Ap), apigenin (Ag), gallic acid (GAE), fresh weight (fw) and dry weight (dw).

## Data Availability

Not applicable.

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
