# Peer review of "Analysis of the Qualitative and Quantitative Content of the Phenolic Compounds of Selected Moss Species under NaCl Stress"

_molecules, 2023, doi:10.3390/molecules28041794_

Round 1

Reviewer 1 Report

In attachment

Author Response

Thank You very much for You constructive comments and improvement suggestions. We appreciate it very much.

Here are the point to point answers to Your comments:

Authors of the submitted manuscript preformed observational study and report on changes in the content of 9 phenolic compounds in 3 mosses: Physcomitrium patens, Entosthodon hungaricus and Hennediella heimii exposed to salt stress. It was found that each moss responded in species-specific and salt concentration-specific manner regarding the presence of these phenolic compounds. Authors concluded that different biochemical mechanisms are involved in responses of these species.

Authors examined the presence of chosen phenolic compounds, but without determining total phenolic content in each experimental case.

On the contrary, these are present in Fig. 1A, Fig. 2A, and Fig. 3A, both in short- and long-term stresses for each tested species. Thus we have it already incorporated within the present manuscript! Please refer to the figures.

That information should be included. Already there.

Further, at least one HPLC spectrum for each species should be given (for example, without salt stress) in order to document how many phenolic compounds can be detected in extracts (spectra could be given as supplementary material). From the data reported, it can’t be concluded why only these 9 compounds were investigated — no others, too small concentrations or some other reason?

Thanks for these comments. Of course, there can be other phenolic compounds found in mosses. However, based on previous tests in our lab and the references, we decide to compare the nine represented with the highest peaks and present in all tested species. Some others were irrelevant and/or undetectable to us, due to too small concentrations. 

In Results section, authors propose joint activity of different mechanisms in combating salt stress. Elaborate this statement, suggest joint mechanisms.

According to the results present in literature for vascular plants, tolerance mechanisms include also some other physiological and biochemical responses of plants. These experimental results however are largely missing for bryophytes, thus based on related organisms, our ongoing experiments, and rare literature sources, we expect phenolics not to be the only level of defense in salt caused stress. There are some papers suggesting from the molecular studies (DNA/RNA), the presence of many other osmoprotective compounds detected in model moss P. patens (e.g. reference in our manuscript  - 6, 7, 8, 17, 40, 43). However, we are not aware that these were tested experimentally.

Can you put the obtained results in real context? What is the maximum concentration of salt that can be expected in nature, thus what is the most likely phenolic profile that can be expected as a response to salt stress in nature?

A very good point. Thanks for that comments.

Mosses are small organisms, and with very small growth. Thus, the analyses of chemical constituents are limited to those that can be collected in huge biomass. Therefore, such results from the real context are widely missing. On the other hand there are too many factors in real habitats affecting the chemical content that the study like ours is missing to give us any insight into obscure mechanisms of salt stress and resistance in mosses. The light, drought or even temperature can act synergistically or antagonistically and blurred real results. Thus lab experimental step by step work is rather welcomed. Thus maximal concentration is rather environmental depending. Additionally, other salts can increase/decrease or change the tolerance in various species. This is elaborated in reference 2 and stated in our manuscript.

Also, we add the additional comments in discussion chapter.

In Discussion, authors stated (lines 195-198):               phenolics decreased heavily when plants were exposed to extremely high concentrations of NaCI suggesting that phenolics are the main component that contribute to salt tolerance due to their degradation and low concentration.” No experiment was performed to document degradation, so that can’t be stated. Also, change in other compounds was not assessed here, so it can’t be said that phenolics are “the main component” in salt tolerance.

Thanks for the comments. We agree. We have changed it, and we did avoid the word - degradation. 

It was also stated (lines 223-226): “... suggest the important role of this specific phenolic acid in the response to NaCI.” Only the presence was investigated, not the role.

Thanks for the comments. We agree. We have changed it, to avoid the word - role.

There is over-selfcitation of authors — 13/54 references. 25%!

We are the only group dealing with laboratory experiments on non model mosses, i.e. not only on P. patens that apply also stresses. Thus, it was necessary to link this with our previous results, at least for direct comparisons. Even, the reviewers asked to add or link, and we did it where needed. We do also moss biology studies out of the lab. The focus on salt tolerant bryophytes, which is pure habitat related studies, are missing since having no economic importance, they are simply neglected.  Thus, we think it is not over-selfcitations. It does not serve us, and we have much more related references, we could include, but we did it tentatively related to this study. The other reviewers asked to include more details on previous knowledge and these goes back to our previous studies. The two of the studied species are extremely rare in nature, thus rarely experienced by researchers.

Discussion should be cleared from repetition and more focused.

We did change to some extent. Even, You asked to add natural context or native background. There they are.

Minor remarks:

-English needs correction,

Done, British native speaker done the English revision.

-On several occasions species are not in italic

We have checked up. Changed.

-Occasionally, there is repetition of text (e.g. lines 130-133),

Thanks for remark. We have changed it.

-Line 171:          undoubtedly may indicate...” — either “undoubtedly” or “may”,

Thanks for remark. We have changed it.

-“decrement“ — decrease,

Thanks for remark. We have changed it, twice.

-“decreased heavily” — significantly reduced,

Thanks for remark. We have changed it.

-Line 215-216: “Although chemical diversity is rather more but insufficiently studied...” — rephrase,

Thanks for remark. Done.

-Unify units, e.g. g/L (given for sucrose concentration) or mol/L (given for salt concentration),

Thanks for remark. Done.

-In section 4.2, authors talk about ethanol extract, in section 4.3 about methanol extract!

Thanks for remark. We missed m. Both were methanol extraction.

-In section 4.2, what was the quantity of sodium carbonate added and in what form (solid, solution)?

Thanks for remark. Added.

-Check references in text/reference list (e.g. 50 seems to be given to two different papers),

Thanks for remark. Redone.

-Significant difference should be at p < 0.05 (not p fi 0.05).

Thanks for remark. Redone.

Reviewer 2 Report

"The obtained results clearly  demonstrate that phenolics are included in salt and oxidative stress responses both in halophytic and non-halophytic mosses and also suggest that the mechanisms are different among tested mosses." 

Dear authors, the final conclusion does not appear either in the discussion or in the presentation of your results. it will be useful to demonstrate your conclusion well in the discussion part. Although you mention that the results were statistically valid in certain analysis is not clear how mass spectroscopy can provide accurate quantification on phenolics. You are requested to argue

Author Response

Many thanks for the comments. 

The suggestions are included in the text of discussion part and marked in red.

The methodology used allows us the quantification on phenolics in rather accurate way, since these is proven to be adequate in many studies (e.g. Molecules. 2013 Sep; 18(9): 10213–10227. doi: 10.3390/molecules180910213).

Reviewer 3 Report

Review

 The selected moss analyses of the qualitative and quantitative con-2 tent of phenolic compounds under NaCl stress

This manuscript is describing an interesting topic about salt stress and phenolic compound induction using in vitro mosses cultures but needs serious improvement.

The entire manuscript is unpolished, rushed in results and discussion, missing key elements in section material and methods, and depth in discussion and conclusion sections.

Langue needs serious improvement and many mistake corrections.

 Figures need to be remade. In the mosses picture, nothing is visible, they need to be completely changed.

 Bars in figures are impossible to read and all Figures used statistical specifications are missing. Please correct all this.

 Please write Latin names in Italic script e.g. Line 108.

Fig .1-3  plot C and D is impossible to read in this color.

The Abstract:

 Needs to be re-written, as it is now not showing any results obtained as well as grammar mistakes that need to be corrected:

Line 17…, and

Line 18… were salts

Line 19 …as a non-enzymatic

Line 21 … to be a common

etc.

Keywords: sodium chloride exchange to in vitro cultures

 Introduction:

Poorly written, and needs improvement. Also, correct all grammar mistakes.  Please add specific information about the mosses sp. you are working with as well as more deep reasons why this work has been conducted and is important. Also, mention here that you have been working with in vitro cultures.

The botanical characterization of used mosses is missing.

Results

Need revision. Please correct and improve all Figures.

Do you have any information about morphological and anatomical changes of in vitro plants (shoots or roots) growing under salt stress? The picture you have in the ms now are not showing any visible differences, please change them.

Discussion

Can be improved, correct grammar in this section and make it more elaborated in depth. 

Line 180, 184  grammar, etc.

No discussion about the source of the tissue used for the analyses.

Material & Methods:

Needs improvement and several corrections.

E.g. the first paragraph Line 276-280 is not needed at all in this section,  remove them.

 Please instead specified where the plant material is coming from, and how in vitro cultures were obtained, sub cultured describe media used for in vitro propagation. Describe in detail the composition of the media used. How often were the cultures subculture, and how old plants were used for the analyses? Which part of the plant in vitro was used for analyses?

Information about the Chemicals suppliers is missing.

Where the in vitro cultures were growing, in the growth chamber, and which type?   Did they have roots?

In section 4.2 specifies in detail which part of in vitro plants was used for the biochemical analyses.

Conclusions

Needs more elaboration in-depth.

References

A lot of old literature is used in this ms. For the first reference write just the first author and follow et al.

Thank you.

20.11. 2023

Reviewer 4 Report

This manuscript entitled "The selected moss analyses of qualitative and quantitative con- 2 tent of phenolic compounds under NaCl stress" was written in a sound English language that is almost free from linguistic and scientific errors and studied the effect of salt stress on one non-halophytic (funaroid Physcomitrium patens) and two halophytic mosses (funaroid Entosthodon hungaricus and pottioid Hennediella heimii) .

The authors used relevant references in the research point, which added a distinctive value to the manuscript's content and the state of the art of the scientific discipline in this field. The manuscript included 54 references, including 11 modern research published during the last five years, with 2 most recent references published during this year 2022.

The introduction:

Using about 28 references, the introduction reviewed the bryophytes background and the environmental and economic importance of its species. It also listed the importance and presence of antioxidants in bryophytes.

Results and Discussion:

The discussion of the results was carried out correctly in 3 main figures. Sub-divided each of them into 4 graphs showing the total phenolic, content of phenolic compounds, and relative representation of phenolic compounds in both short and long-term stresses.

Materials and Methods:

This section was written very clearly and used detailed methods for extraction and identification of total phenols, as well as astragalin, isoquercitrin, rutin, p-coumaric acid, caffeic acid, kaempferol, naringenin, apigetrin, and apigenin in the three studied mosses using UHPLC/-HESI-MS/MS techniques

The final decision is to accept this manuscript for publishing as it is.

Author Response

Thank You very much indeed.

Round 2

Reviewer 1 Report

Authors have taken into consideration comments and suggestions, and revised the manuscript accordingly.

Author Response

Thank You for your response. 

Manuscript has been revised by Native English speaker - linguist. Therefore, minor English changes has already been done.

Thank You again for constructive suggestions and comments.

Reviewer 3 Report

2nd Review

The selected moss analyses of the qualitative and quantitative con-2 tent of phenolic compounds under NaCl stress.

The English language has been improved and many grammar mistakes were corrected in the revised version of ms which shows that the submission of the original version of this document has been rushed in the first place.

Many of the comments were not accepted by the authors which is a shame because the revised version still needs improvement to be accepted for publishing in Molecules.

Figures were not changed nor improved which definitely needs to be done to meet the standards of this journal.

Pictures of mosses in all three figures are in bad resolution and the size is very small.  Pictures are visible with the soft edges and most probably they were prepared for the PowerPoint presentation and not for publication.

The discussion in this version of ms is not more elaborate as suggested which is a pity.

The section Material and Methods was not elaborated on and the request to describe the protocol used for growing in vitro cultures with more technical details has been denied. The reader of this manuscript is not obliged to look for previously published work if they want to know the technical details of the work presented here. This request is very relevant because this was the type of plant material you used for your analyses.

 And BTW out of 54 references, only 7 are from the years 2019 to 2022, and out of them, 4 are self-citations.

The accusation that I do not like this ms is not relevant.  I would not spend my valuable time reading you ms twice if I would not consider it interesting.

The current version of the ms can still be improved and I can not recommend it for publishing in Molecules in the present stage.

1.2. 2023

Author Response

2nd Review

The selected moss analyses of the qualitative and quantitative con-2 tent of phenolic compounds under NaCl stress.

The English language has been improved and many grammar mistakes were corrected in the revised version of ms which shows that the submission of the original version of this document has been rushed in the first place.

Thank You for noticing. We tried to improve it according to Your suggestions. Still, You have tick to minor English changes in the Review Form. Please, suggest which ones You have in mind.

Many of the comments were not accepted by the authors which is a shame because the revised version still needs improvement to be accepted for publishing in Molecules.

We made changes according to direct and clear suggestions. However, other instructions were not clearly stated even though we would like to make improvements of the ms.

Figures were not changed nor improved which definitely needs to be done to meet the standards of this journal.

We have already addressed this issue in previous reply. Here is the copy of it. In summary, photographs are incorporated only to represent the moss species that figures are referring to.

Bars in figures are impossible to read and all Figures used statistical specifications are missing. Please correct all this.

Sorry, but this is not true. Please, refer to the figure again. In 1A, 2A, and 3A, there are in the graphs. Figures 1B, 2B, and 3B do not undergo statistical error analyses since these are representations of summary measurements for each different phenolic compound combined in one bar.”

Pictures of mosses in all three figures are in bad resolution and the size is very small.  Pictures are visible with the soft edges and most probably they were prepared for the PowerPoint presentation and not for publication.

We have changed the images of mosses, but we can remove them completely from figures if it is necessary.

The discussion in this version of ms is not more elaborate as suggested which is a pity.

We did it in the previous version and marked changes in red. However, You have not stated exactly what are You referring to. The other changes are already incorporated into last version and/or replied to You in point-to-point.

The section Material and Methods was not elaborated on and the request to describe the protocol used for growing in vitro cultures with more technical details has been denied. The reader of this manuscript is not obliged to look for previously published work if they want to know the technical details of the work presented here. This request is very relevant because this was the type of plant material you used for your analyses.

We do not agree with this, since this is unnecessary repetition. However, we did as suggested, i.e. types of the mineral salts used for BCD medium were named as well as their concentrations used. Protocols for media preparations are now incorporated in MM section.

And BTW out of 54 references, only 7 are from the years 2019 to 2022, and out of them, 4 are self-citations.

Yes, indeed, if we compare 2020 to 2022, even less/more. Why is this important? Is there any reason to pay attention for the year of reference, if the references bear the relevant data up to date?

The accusation that I do not like this ms is not relevant.  I would not spend my valuable time reading you ms twice if I would not consider it interesting.

The current version of the ms can still be improved and I can not recommend it for publishing in Molecules in the present stage.

1.2. 2023